# Recording of intellectual disability in general hospitals in England 2006–2019: Cohort study using linked datasets

Rory Sheehan[1]*, Hassan Mansour[2], Matthew Broadbent[1], Angela Hassiotis[2], Christoph Mueller[1], Robert Stewart[1,3], Andre Strydom[1], Andrew Sommerlad[2]

1 Institute of Psychiatry, Psychology and Neuroscience, King's College London, London, United Kingdom, 2 Division of Psychiatry, University College London, London, United Kingdom, 3 South London and Maudsley NHS Foundation Trust, London, United Kingdom

* rory.sheehan@kcl.ac.uk

**Data Availability Statement:** The data used in this work have been obtained from the Clinical Record Interactive Search (CRIS), a system that has been developed for use within the NIHR Mental Health

## Abstract

### Background

Accurate recognition and recording of intellectual disability in those who are admitted to general hospitals is necessary for making reasonable adjustments, ensuring equitable access, and monitoring quality of care. In this study, we determined the rate of recording of intellectual disability in those with the condition who were admitted to hospital and factors associated with the condition being unrecorded.

### Methods and findings

Retrospective cohort study using 2 linked datasets of routinely collected clinical data in England. We identified adults with diagnosed intellectual disability in a large secondary mental healthcare database and used general hospital records to investigate recording of intellectual disability when people were admitted to general hospitals between 2006 and 2019. Trends over time and factors associated with intellectual disability being unrecorded were investigated. We obtained data on 2,477 adults with intellectual disability who were admitted to a general hospital in England at least once during the study period (total number of admissions = 27,314; median number of admissions = 5). People with intellectual disability were accurately recorded as having the condition during 2.9% (95% CI 2.7% to 3.1%) of their admissions. Broadening the criteria to include a nonspecific code of learning difficulty increased recording to 27.7% (95% CI 27.2% to 28.3%) of all admissions. In analyses adjusted for age, sex, ethnicity, and socioeconomic deprivation, having a mild intellectual disability and being married were associated with increased odds of the intellectual disability being unrecorded in hospital records. We had no measure of quality of hospital care received and could not relate this to the presence or absence of a record of intellectual disability in the patient record.

### Conclusions

Recognition and recording of intellectual disability in adults admitted to English general hospitals needs to be improved. Staff awareness training, screening at the point of admission,

Biomedical Research Centre (BRC) at the South London and Maudsley NHS Foundation Trust (SLaM), and Hospital Episode Statistics (HES), a dataset collated by NHS Digital. Individual-level data are restricted in accordance with patient led governance established at SLaM, and by NHS Digital in the case of linked HES data. CRIS data are available for authorised researchers who meet the criteria for access as (1) SLaM employees or (2) those having an honorary contract or letter of access from SLaM. Further details can be obtained by e-mailing cris.administrator@kcl.ac.uk.

**Funding:** The authors received no specific funding for this work. AS has a research fellowship from the Wellcome Trust [222932/Z/21/Z]. The funder had no role in the study design; collection, analysis, and interpretation of data; writing of the paper; and/or decision to submit for publication.

**Competing interests:** I have read the journal's policy and the authors of this manuscript have the following competing interests: RSt declares research support received in the last 36 months from Janssen, GSK, and Takeda, and royalties received from Oxford University Press.

**Abbreviations:** aOR, adjusted odds ratio; BRC, Biomedical Research Centre; CIPOLD, Confidential Inquiry into Premature Deaths of People with Learning Disabilities; CLDT, community learning disability team; COVID-19, Coronavirus Disease 2019; CRIS, Clinical Record Interactive Search; EHR, electronic health record; GATE, General Architecture for Text Engineering; HES, Hospital Episode Statistics; HoNOS-LD, Health of the Nation Outcome Scales for People with Learning Disabilities; IMD, Index of Multiple Deprivation; NHS, National Health Service; NLP, natural language processing.

and data sharing between health and social care services could improve care for people with intellectual disability.

## Author summary

### Why was this study done?

- People with intellectual disability have specific health needs and are more likely to be admitted to hospital than people without intellectual disability.

- Existing evidence shows that hospital care for people with intellectual disability needs to be improved if the worse health outcomes and health inequalities experienced by this group are to be overcome.

- Intellectual disability needs to be recognised in healthcare settings in order that individual reasonable adjustments to care can be made.

### What did the researchers do and find?

- We joined 2 databases covering from 2006 to 2019 to investigate how often intellectual disability is recorded when people with the condition are admitted to general hospitals in England.

- We included 2,477 people with intellectual disability who together had 27,314 admissions to hospital.

- Intellectual disability was accurately recorded in only 2.9% of all admissions and was more likely to be unrecorded in people with less severe intellectual disability and those who were married.

### What do these findings mean?

- Recognition and recording of intellectual disability in general hospitals in England needs to be improved and must then be followed by processes and procedures that impact the quality and effectiveness of care that people receive.

- Using English Hospital Episode Statistics (HES) data alone to identify people with intellectual disability accessing general hospital care is insufficient and data linkage with other sources is necessary to obtain a more accurate picture for research and service planning purposes.

## Introduction

Intellectual disability is a lifelong disorder characterised by deficits in general cognitive ability and impaired functional skills [1]. People with intellectual disability constitute between 1 and 2% of the population, equating to approximately 1 million people in England alone [2]. Adults with intellectual disability have worse physical and mental health than those without

intellectual disability, including higher rates of long-term conditions and complex multi-morbidity [3–5], and die up to 20 years younger than the general population [6,7]. This mortality gap is consistent across high-income countries [8]. Furthermore, roughly one-third of deaths of people with intellectual disability are potentially avoidable with the provision of good quality healthcare [6,9–11]. Addressing the health inequalities experienced by this group is a priority for governments in the United Kingdom (UK) and beyond [12,13].

People with intellectual disability are more likely to be admitted to general hospitals, where they stay longer than those without intellectual disability [14,15]. They are at risk of receiving poor quality care and of their needs not being met for reasons that include: lack of knowledge and understanding among health professionals, diagnostic overshadowing, and institutional discrimination [16–18]. People with intellectual disability and their carers frequently report poor experiences of general hospital care, including inadequate communication and failure to acknowledge carers' expertise [19–21].

Recognition of intellectual disability is essential to allow additional support needs to be identified and reasonable adjustments for these. Healthcare for All, a UK government-funded inquiry into access to healthcare for people with intellectual disability, highlighted the need to identify people with intellectual disability at all points of healthcare delivery, including hospital admission [17]. However, little information exists on how well intellectual disability is recognised and recorded in general hospital settings in England. One existing study suggests poor recording of intellectual disability in people who are admitted to hospital but was based on ecological data which is at risk of bias [22].

In this study, we sought to:

1. investigate the recording of intellectual disability in adults with a confirmed intellectual disability diagnosis who were admitted to general hospitals;

2. analyse changes in recording of intellectual disability in those admitted to hospital over time; and

3. identify clinical and sociodemographic factors associated with intellectual disability being unrecorded in those with the condition.

## Methods

### Study setting

This study was conducted in England where most healthcare is provided by the National Health Service (NHS), a state-funded provider. Secondary (specialist) healthcare is delivered by organisations known as Trusts. Acute Trusts provide general hospital services for physical health problems, including in-patient and out-patient facilities, emergency departments, and surgical care. Mental health Trusts provide secondary mental healthcare including services for the assessment, diagnosis, and psychiatric management of people with intellectual disability.

The South London and Maudsley (SLaM) NHS Trust is one of the largest providers of secondary mental healthcare in Europe, serving a population of approximately 1.2 million people distributed between 4 demographically diverse south London boroughs. SLaM's comprehensive mental healthcare services include those for diagnosis and psychiatric management of people with intellectual disability, which are provided by dedicated multidisciplinary community learning disability teams (CLDTs) in each borough.

### Study design and data source

This is a retrospective cohort study using data from 2 linked clinical datasets, the SLaM Biomedical Research Centre (BRC) case register and the English Hospital Episode Statistics (HES)

database. The study protocol document, written before the data were extracted and any analysis was performed, is included as a supplementary file (S1 Text).

## South London and Maudsley Biomedical Research Centre case register

The SLaM BRC case register contains deidentified electronic health records (EHRs) of over 500,000 patients who have received care from any SLaM service since April 2006 [23]. Data are recorded either in structured fields (e.g., age, ethnicity, and diagnosis) or as part of the unstructured free text record consisting of correspondence and other clinical case notes. The Clinical Record Interactive Search (CRIS) software was created to enable users to extract demographic and clinical information from the EHR for scientific research [23]. CRIS deploys over 100 natural language processing (NLP) algorithms developed on General Architecture for Text Engineering (GATE) software [24], developed over the last 10+ years, to extract relevant information from the free text record [25], in addition to data from structured fields.

## Hospital Episode Statistics database

HES is a national dataset compiled by NHS general hospital providers, and curated by NHS Digital, that includes details of all in-patient admissions, out-patient appointments, and attendances to emergency departments in England [26]. HES data are used to monitor activity and as the basis for remunerating hospitals for the care they provide; they can also be used for secondary research in a fully anonymised format. We used HES in-patient admission data that include diagnoses identified during the hospital contact which have been added to the in-patient discharge summary, recorded using codes of the International Statistical Classification of Diseases and Related Health Problems, 10th edition (ICD-10) [27]. Up to 20 diagnostic codes can be added for each patient's HES record. We also obtained admission and discharge dates, and admission route (elective/planned or non-elective/emergency).

## Study participants

We retrieved records of all adults ($\geq$18 years) in the SLaM BRC case register with a diagnosis of intellectual disability who had received care from SLaM NHS Trust between 2006 and 2019; this timeframe was selected to maximise the data available for analysis and in order that longitudinal trends could be investigated, finishing prior to the Coronavirus Disease 2019 (COVID-19) pandemic which greatly impacted in-patient healthcare. A diagnosis of intellectual disability was taken as either a record of an ICD-10 code within the mental retardation subchapter (ICD-10 code F70 to F79) or a diagnosis of intellectual disability in the free text, extracted using the relevant NLP algorithm. These individuals' patient records were linked with HES data over the same period using approved secure processes via the SLaM Clinical Data Linkage Service [28] to identify all general hospital in-patient admissions during the study period and diagnoses recorded within the general hospital setting during each admission. We did not measure attendances to the out-patient department or visits to the emergency department which did not result in an admission.

## Co-variates

The following data were extracted from the structured fields of the SLaM BRC case register for each participant using the recording closest to their first general hospital admission: age, sex, ethnicity (white, Asian, black, mixed, and other), marital status (married, civil partnership, cohabiting, single, divorced, separated, and widowed). Degree of intellectual disability (mild, moderate, severe, and profound) was extracted first from the patient's latest ICD-10 diagnostic

code (where the second character denotes level of disability) or, if this was not specified, using the Health of the Nation Outcome Scales for People with Learning Disabilities (HoNOS-LD), an outcome measure recommended for routine clinical use with this population [29], or from free text records. Socioeconomic status of participants was estimated using the Index of Multiple Deprivation (IMD), a widely used neighbourhood-level (each area comprising approximately 1,500 individuals) measure of relative deprivation based on 37 indicators related to the patient's address [30].

## Analysis

The BRC case register record of intellectual disability was used as the "gold standard" diagnosis against which recording of intellectual disability in general hospitals was tested. The intellectual disability services within SLaM that contribute to the BRC register specialise in diagnosis and management of people with intellectual disability. It is standard procedure to add these diagnoses to the record using a structured electronic form. Diagnosis of intellectual disability is by trained professionals who are experienced in using standard classification systems and is expected to be made following a combination of formal cognitive testing, assessment of adaptive functioning, and evidence that deficits have been present since at least childhood, so we judged that the specialist secondary mental health service would be an appropriate gold standard.

Summary statistics were used to describe the sample. We calculated proportion of people or episodes that are correctly identified as having an intellectual disability in general hospital records (HES) after the first recorded diagnosis in the BRC case register, which we will hereafter refer to as "sensitivity":

a. for each admission (proportion of all admissions of people with intellectual disability during which intellectual disability is recorded) (admission sensitivity);

b. for each patient (proportion of people with intellectual disability who have the diagnosis recorded in any hospital admission) (patient-level sensitivity);

c. for emergency admissions only (proportion of all emergency (non-elective) admissions of people with intellectual disability in which intellectual disability is recorded), as the majority of elective admissions are only brief and recurrent admissions, e.g., for renal dialysis or wound dressing during which we judged full diagnostic assessment may not be appropriate or customary.

For each of these we investigated intellectual disability recording in the HES data using:

i. ICD-10 intellectual disability codes (F70 to F79);

ii. ICD-10 intellectual disability codes (F70 to F79) and a single additional code, F81.9 (developmental disorder of scholastic skills, unspecified), which we noted during initial exploration of the data was used in a significant proportion of people with a BRC case register diagnosis;

iii. ICD-10 intellectual disability codes (F70 to F79), the F81.9 code, and 48 individual ICD-10 codes for specific disorders almost always associated with intellectual disability from a code list developed previously (Supporting information, S1 Table) [31].

We investigated time trends in recording of intellectual disability in general hospitals by reporting the proportion of each individual's first emergency hospital admission between 2006 and 2019 in which intellectual disability was recorded. We reported on the trend using chi-squared test for trend.

We used logistic regression to identify factors associated with intellectual disability (F70 to F79) being unrecorded in HES data. Univariate regression was conducted for each variable followed by multivariable analysis adjusted for each co-variate and number of hospital admissions in the study period, categorised as 1; 2–10; >10. Marital status was collapsed into a binary variable for this stage of the analysis (married or unmarried) in response to small numbers in some response categories. In a sensitivity analysis, we used multiple imputation by chained equations to impute missing values for all variables that contained missing data [32] and conducted logistic regression on each of 20 imputed datasets, combining coefficients using Rubin's rules [33]. All analysis was performed using STATA v14.

As a sensitivity analysis, we examined factors associated with unrecorded diagnosis during each admission. As these admission-level data have a multilevel data structure whereby many patients had several admissions with correlation between these likely, we used a mixed effects logistic regression with a random effect for intercept meaning that the odds ratios derived from the analysis reflect the risk of diagnosis being unrecorded during each admission according to each patient characteristic. We then repeated these models using multiple imputation to impute missing sociodemographic or clinical characteristics.

## Ethics

The SLaM BRC case register and CRIS have received ethical approval from the Oxfordshire Research Ethics Committee C (18/SC/0372) for secondary analysis of deidentified health data. Researchers did not have access to patient-identifiable information.

This study is reported as per the REporting of studies Conducted using Observational Routinely collected Data (RECORD) guideline (S1 Checklist).

## Changes from originally planned work

We had planned to also investigate recording of autism as part of this work but restricted our analysis to intellectual disability to make for a more focused paper and coherent narrative. Some changes were made to the protocol after further discussion within the research team but before data extraction or analysis. We extended the date of patient identification to 2019 to permit a larger sample size and so that time trends would become more apparent. We had planned to only include data from emergency admissions but chose to also include elective admission data and to report this separately in the analysis in order to gain further insights into rates of recording. We chose not to include Health of the Nation Outcome Scales (HoNOS) data (a clinician-rated measure of psychiatric symptoms and social functioning) owing to high amounts of missing data and because the scale is not designed to capture the full range of potential impairments relevant to people with intellectual disability. We had planned to analyse whether recording of intellectual disability was associated with time to admission. However, after further consideration, we felt that time between admissions is not a valid proxy for quality of care received during an in-patient admission, as people with intellectual disability may have complex conditions that necessitate regular admissions, irrespective of quality of care, and the frequency of admissions is likely to be related to many other factors, including the quality of community health and social care.

After the data extraction, we noted the high rates of use of the F81.9 diagnostic code; this was not expected but was felt to be relevant to understanding coding in a clinical context, and therefore, these codes were added to subsequent analyses while also presenting findings without this code.

In response to comments of peer-reviewers, we undertook additional analyses which are presented in the final version of the paper. At this stage, we updated the analysis to include a

list of ICD-10 codes associated with specific causes of intellectual disability. We also included the sensitivity analysis following comments of reviewers, examining factors associated with diagnostic recording using mixed effects logistic regression.

## Results

### Sample characteristics

The sample comprised 2,477 adults with intellectual disability who were admitted to a general hospital in England at least once over the course of the study period. Details of the sample are provided in Table 1. There was a slight preponderance of males (53.9%) and the majority had mild intellectual disability (59.1% with data available). The average age at first general hospital admission was 44 years. The largest ethnic group was white. Most (83.5%) were unmarried.

There were 27,314 discrete admissions to general hospitals over the study period; 16,270 were non-elective admissions and the remainder (11,044) were elective admissions (e.g., for planned surgery, routine dialysis, or change of wound dressing). The median number of total admissions per patient was 5.

### Sensitivity of general hospital recording of intellectual disability

Taking each of the 27,314 admissions independently, 788 had an HES record of intellectual disability (F70 to F79) (admission-level sensitivity = 2.9%, 95% CI = 2.7, 3.1). Of each

**Table 1. Demographics of adults with diagnosed intellectual disability admitted to an English general hospital during the study period (*n* = 2,477).**

| | | *n* | % |
|---|---|---|---|
| **Age** | Mean (SD) | 44.0 (16.1) | - |
| **Degree of intellectual disability** | Mild | 928 | 37.5 |
| | Moderate | 420 | 17.0 |
| | Severe | 208 | 8.4 |
| | Profound | 13 | 0.5 |
| | Missing | 908 | 36.7 |
| **Sex** | Male | 1,335 | 53.9 |
| | Female | 1,142 | 46.1 |
| **Ethnicity** | White | 1,517 | 61.2 |
| | Asian | 114 | 4.6 |
| | Black | 539 | 21.8 |
| | Mixed | 73 | 3.0 |
| | Other | 56 | 2.3 |
| | Missing | 178 | 7.2 |
| **Marital status*** | Married | 112 | 4.5 |
| | Unmarried | 2,068 | 83.5 |
| | Missing | 297 | 12.0 |
| **Deprivation score**** | Mean (SD) | 29.1 (10.7) | - |
| | Missing | 71 | - |
| **Number of admissions** | Range | 1–740 | - |
| | Median (IQR) | 5 (3–10) | - |

*Married includes civil partner, co-habiting; unmarried includes single, widowed, and divorced.

**Deprivation score is the IMD. Higher scores indicate a greater degree of deprivation.

IMD, Index of Multiple Deprivation; IQR, inter-quartile range; SD, standard deviation.

**Table 2. Sensitivity of recording of intellectual disability in English general hospital records 2006–2019 for individual admissions and for individual patients by ICD-10 code group.**

| ICD-10 descriptor and codes | Sensitivity (95% CI) | | |
|---|---|---|---|
| | Intellectual disability (F70–F79) | Intellectual disability (F70–F79) Developmental disorder of scholastic skills, unspecified (F81.9) | Intellectual disability (F70–F79) Developmental disorder of scholastic skills, unspecified (F81.9) Disorders associated with intellectual disability (*) |
| For each admission (admission sensitivity) | 2.9 (2.7, 3.1) | 27.7 (27.2, 28.3) | 30.6 (30.1, 31.2) |
| For each patient (patient-level sensitivity) | 18.0 (16.5, 19.5) | 66.3 (64.4, 68.1) | 69.2 (67.4, 71.1) |
| For each admission (emergency admissions only) | 2.0 (1.8, 2.2) | 32.7 (32.0, 33.4) | 36.0 (35.2, 36.7) |

*See Supporting information (S1 Table) for a full list of codes of disorders associated with intellectual disability.

emergency admission ($n$ = 16,270), 319 had a record of intellectual disability (sensitivity = 2.0%, 95% CI 1.8, 2.2). Of the 2,477 people who were admitted to hospital, 445 had a record of intellectual disability defined using intellectual disability codes (F70 to F79) at any time in their general hospital record (patient-level sensitivity = 18.0%, 95% CI = 16.5, 19.5).

Including the ICD-10 code F81.9 (developmental disorder of scholastic skills, unspecified) in addition to the F70 to F79 codes resulted in a notable increase in those with intellectual disability who were recorded in the general hospital record (admission-level sensitivity = 27.7% (27.2, 28.3); patient-level sensitivity = 66.3% (64.4, 68.1)) (Table 2). Adding a list of codes for specific disorders associated with intellectual disability resulted in a small increase in the proportion of admissions in which intellectual disability was recorded (admission-level sensitivity = 30.6% (30.1, 31.2); patient-level sensitivity = 69.2% (67.4, 71.1)).

## Time trends in recording intellectual disability in general hospital records

Data for recording of intellectual disability stratified by year for the first emergency admission of each patient are shown in Fig 1. Strict recording of intellectual disability using F70 to F79 codes showed little overall change over the study period. Including the F81.9 code with F70 to F79 codes showed a consistent increase in those who were identified, from 17.5% (95% CI = 13.8, 21.7) in 2005 to 62.5% (40.6, 81.2) in 2019 ($\chi^2$ for trend = 138.7, $p < 0.001$). Raw data and proportions are given in Supporting information (S2 Table).

## Associations with intellectual disability being unrecorded in hospital records

Factors associated with a person with intellectual disability never having this accurately recorded (as ICD-10 codes F70 to F79) in their general hospital record were investigated (Table 3). In the adjusted analysis, having more severe intellectual disability was associated with lower odds of intellectual disability being unrecorded (adjusted odds ratio (aOR) for severe versus mild intellectual disability 0.30 (0.20, 0.46), $p < 0.001$), and being married was associated with higher odds of intellectual disability being unrecorded (aOR 3.12, 95% CI 1.10, 8.81, $p$ = 0.03). Regression results with imputed values were similar and are given in Supporting information (S3 Table).

The results of the sensitivity analysis using the mixed effects regression model are presented in Supporting information (S4 and S5 Tables). In common with the logistic regression, the odds of intellectual disability being unrecorded during each hospital admission were lower in

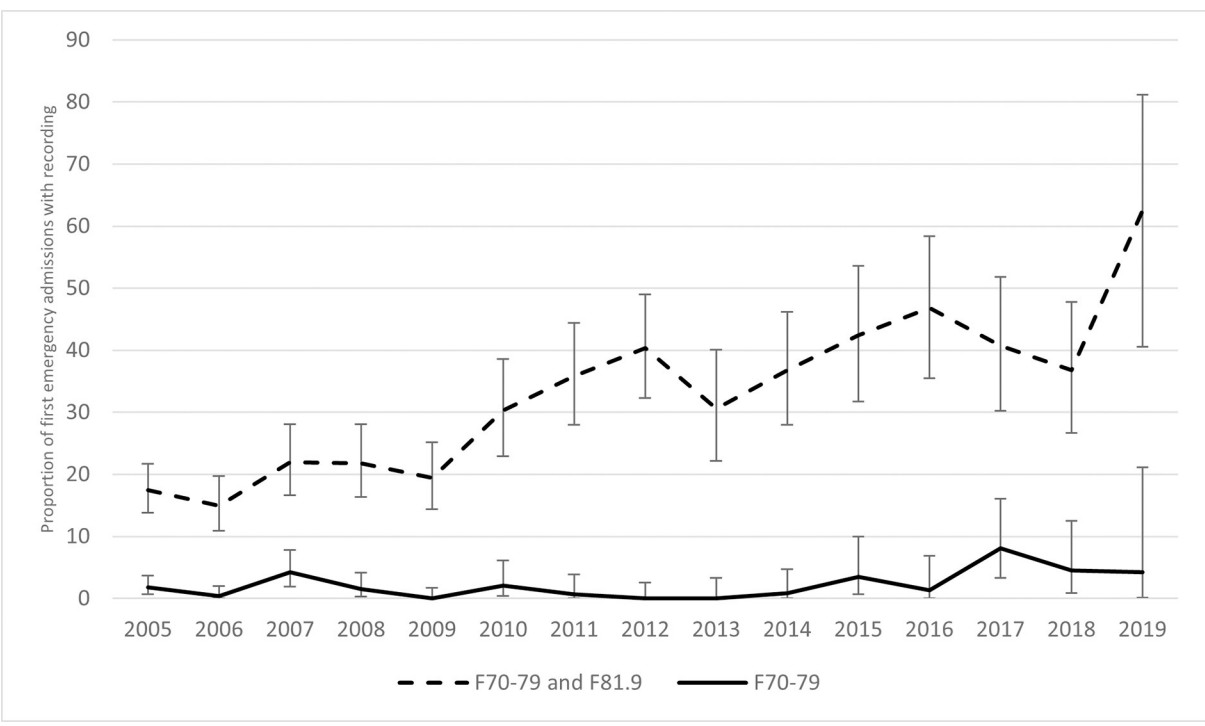

**Fig 1. Time trends in recording of intellectual disability in those admitted to a general hospital in England, 2005–2019.** F70–F79, codes for intellectual disability; F81.9, additional nonspecific code for developmental disorder of scholastic skills, unspecified. Error bars represent 95% confidence intervals.

those with a greater degree of intellectual disability and higher in those who were married. Being younger and being of Asian ethnic background were independently associated with lower odds of intellectual disability being unrecorded and a higher deprivation index was associated with higher odds of intellectual disability being unrecorded. Unrecorded intellectual disability was almost twice as likely in elective compared with emergency admissions.

## Discussion

Accurate recognition and recording of intellectual disability in people who are admitted to general hospitals is important so that additional support needs can be identified and the necessary adaptations to care and processes can be provided. These might include communication support, use of the Mental Capacity Act [34], involving family or paid carers, and addressing issues around planning safe discharge [35]. Collecting and recording accurate data is also important on a population level to allow healthcare providers and commissioners to understand their patient group, allocate adequate resource, and plan services [17].

Using large, linked datasets of real-world clinical data, we found that intellectual disability was poorly recorded in people who were admitted to general hospitals. Taking each admission individually, intellectual disability was accurately recorded in under 3% of all general hospital admissions and just under one-fifth of those with a confirmed diagnosis of intellectual disability (as recorded by specialist intellectual disability and mental health services) ever had the condition accurately recorded in their general hospital records, despite them having several admissions on average. These findings may reflect poor knowledge and recognition of intellectual disability among general hospital staff, a reluctance to label patients with disabilities, or a lack of understanding of why this is necessary.

**Table 3. Odds of intellectual disability being unrecorded in the general hospital record of adults with intellectual disability attending hospital.**

| | | Univariate analysis | | Adjusted analysis* | |
|---|---|---|---|---|---|
| | | Odds ratio (95% CI) | *p*-value | Odds ratio (95% CI) | *p*-value |
| **Age** | (OR per 10 years older age) | **1.13 (1.05, 1.20)** | **<0.001** | 1.08 (0.98, 1.18) | 0.11 |
| **Sex** | Female (reference) | 1 | - | 1 | - |
| | Male | 1.05 (0.86, 1.30) | 0.61 | 1.28 (0.95, 1.73) | 0.11 |
| **Degree of learning disability** | Mild (reference) | 1 | - | 1 | - |
| | Moderate | **0.68 (0.50, 0.93)** | **0.02** | **0.59 (0.42, 0.83)** | **0.03** |
| | Severe | **0.39 (0.27, 0.55)** | **<0.001** | **0.30 (0.20, 0.46)** | **<0.001** |
| | Profound | 0.49 (0.13, 1.79) | 0.28 | 0.71 (0.14, 3.54) | 0.67 |
| **Ethnicity** | White (reference) | 1 | - | 1 | - |
| | Asian | 1.13 (0.67, 1.90) | 0.65 | 1.23 (0.60, 2.49) | 0.57 |
| | Black | 0.85 (0.67, 1.10) | 0.22 | 1.04 (0.72, 1.50) | 0.86 |
| | Mixed | 0.98 (0.53, 1.81) | 0.94 | 1.85 (0.63, 5.42) | 0.26 |
| | Other | 0.58 (0.32, 1.96) | 0.88 | 0.79 (0.32, 1.93) | 0.60 |
| **Marital status** | Unmarried (reference) | 1 | - | 1 | - |
| | Married | **2.12 (1.13, 3.99)** | **0.02** | **3.12 (1.10, 8.81)** | **0.03** |
| **Deprivation index** | (OR per decile higher deprivation) | 1.01 (0.91, 1.11) | 0.92 | 0.96 (0.84, 1.10) | 0.59 |
| **Number of admissions** | 1 | 1 | - | 1 | - |
| | 2–10 | **0.38 (0.24, 0.59)** | **<0.001** | **0.37 (0.20, 0.70)** | **0.002** |
| | More than 10 | **0.19 (0.12, 0.30)** | **<0.001** | **0.16 (0.09, 0.31)** | **<0.001** |

*Adjustment for all variables in the table. Statistically significant results are marked in bold type.

Low rates of recording may also reflect clinical coding errors. We observed a proportion of adults with intellectual disability who were coded as having learning difficulty using the ICD-10 descriptor "developmental disorder of scholastic skills, unspecified." This diagnostic category is ill-defined, and use is discouraged [27] but it might be applied when there are specific deficits in language and speech development, motor co-ordination, or the development of arithmetic, reading, or spelling proficiency without global intellectual impairment. Regular use of this code suggests that some form of cognitive impairment was more frequently recognised during the admission; in practice, this code seems to be used as a proxy for intellectual disability or it may be used when intellectual disability is suspected but not confirmed. Alternatively, it could be used for other unspecified cognitive impairments or borderline intellectual functioning, which does not have a separate category. However, even including the additional code, one third of people with intellectual disability never had a relevant diagnosis recorded, and these diagnoses were unrecorded in around three-quarters of admissions. These codes are also likely to be missed in national administrative returns, leading to underreporting of healthcare use by people with intellectual disability. From a research perspective, using a strict definition of intellectual disability (F70 to F79) in future studies using HES will miss a substantial proportion of people who likely should be included.

Adding codes associated with specific disorders usually associated with intellectual disability [31] increased recording marginally, but the more generic, overarching intellectual disability code may still be needed to identify the person to health systems; for example, those assigned only a specific disorder code in primary care did not were not automatically included in an annual health check scheme designed to identify health needs [36].

There is no English benchmark against which the results of this study can be directly compared, although data suggest that low levels of recording of intellectual disability in healthcare records is an international issue. A recent study conducted in New South Wales, Australia,

found an overall recognition of intellectual disability in only 23.79% of hospital admissions of adults with the condition and that recording had reduced over time [37]. A further Australian study that investigated the recording of intellectual disability in children showed that hospital data identified only 14% those with intellectual disability who had been admitted [38]. Missed recording of intellectual disability is not only a problem in secondary care; far fewer people than would be expected are included on intellectual disability registers held by primary care in England [39]. Recording of intellectual disability within general hospitals is lower than recording of other mental disorders; similar methodology has reported recording rates of 78% for dementia [40] and 56% for schizophrenia [41].

Our data indicate that recording of intellectual disability has improved over time, if the related (but technically incorrect) code of learning difficulty (ICD-10 F81.9) is included. For example, intellectual disability was recorded in over half of new emergency admissions occurring in 2019. The cumulative effect of legislative changes, growing research evidence, and clinical initiatives over the past 15 years may have contributed to increased recognition and recording of intellectual disability in English general hospitals. The Equality Act became law in the United Kingdom in 2010 and strengthened the provisions of existing legislation; this Act not only mandates that service providers make reasonable adjustments to ensure that people are not discriminated against (e.g., by nature of a disability) but also states that providers should be proactive in anticipating and making such adjustments [42]. There is also now a legal duty for public sector providers (including the NHS in England) to publish information to demonstrate compliance with the Act, serving as a further incentive for hospital managers to respond to the needs of disadvantaged groups. The English Confidential Inquiry into Premature Deaths of People with Learning Disabilities (CIPOLD), published in 2013, highlighted the premature mortality and preventable deaths that people with an intellectual disability suffer [6] and prompted the establishment of a national learning disability mortality review programme (known as LeDeR) in 2016. Annual LeDeR reports continue to show significant health inequities [43], thereby maintaining attention on the issue, and stimulating local and national quality improvement initiatives [44]. A relatively new role of learning disability liaison nurse has been established in many acute hospitals in England over the past decade. Learning disability liaison nurses undertake a variety of tasks including frontline patient care, educating staff, influencing strategy, and implementing disability-specific recommendations and guidance [45], all of which are likely to raise the profile of people with intellectual disability throughout the hospital [46]. The trend for improved recording may also reflect overall improvements in diagnostic coding in HES data that has been observed over time [47].

Analysis of factors associated with unrecorded diagnosis showed that having a less severe intellectual disability was independently associated with increased likelihood of intellectual disability never being recorded in general hospital data, in keeping with previous research [37]. Mild intellectual disability may not be immediately obvious on meeting a person with the condition and specific enquiry may not be included in standard medical admission assessments. People with intellectual disability who were married, cohabiting, or in a civil partnership were also more likely not to have their intellectual disability recorded; it may be that assumptions about the lives of people with intellectual disability contributed to this finding, or that these people had milder intellectual disability which was more easily missed, even though we adjusted for intellectual disability severity. However, this finding should be interpreted with caution owing to the relatively small numbers who were married. Other factors investigated were not associated with recording in hospital records in the patient-level analysis. However, in the sensitivity (admission-level) analysis, increasing age at first admission and Asian ethnicity were associated with increased recording of intellectual disability; these positive findings may reflect the greater statistical power from this analysis of over 27,000 hospital

admissions. The relationship with age may also be due to there being more sources of information available to confirm a diagnosis as someone ages (e.g., social care records). The association with ethnicity suggested in this analysis warrants further investigation, but is also based on small numbers and was no longer significant in the imputed analysis. Higher deprivation levels were associated with decreased recording of intellectual disability in the sensitivity analysis; the lack of a consistent association with socioeconomic status may reflect the IMD not being a sufficiently robust measure of deprivation in people with intellectual disability, as many live in group settings which do not reflect the socioeconomic status of themselves or their families [48].

Recording of intellectual disability was better in emergency compared with elective admissions. This is consistent with other studies investigating the recording of dementia and severe mental illness [40,41] and may be due to a more comprehensive medical clerking being undertaken in the emergency department or that emergency admissions are likely to be longer in duration than elective admissions.

## Strengths and limitations

In this study, we were able to identify a large representative cohort using a local specialist mental health and community intellectual disability team case register and link this to a hospital admissions database with national coverage. The results add to the very limited existing data on the recognition and recording of intellectual disability in general hospitals in England and will provide impetus for improvements.

Our study has some limitations. We used diagnosis of intellectual disability made by a secondary mental healthcare service (which includes specialist intellectual disability teams) as the gold-standard meaning that only individuals with intellectual disability living in the catchment area who have accessed secondary mental health and intellectual disability services have been included; those accessing specialist services are more likely to have additional complex needs and may not be representative of the total intellectual disability population. However, our use of specialist service data, interrogation of structured fields, and free-text records using NLP improves confidence in the diagnosis and this approach has been validated in other mental disorders showing high precision [28]. Our approach allowed the construction of a large cohort representative of people with clinically diagnosed intellectual disability, which would not have been possible had we assessed all participants with a standardised assessment.

It is possible that in some cases, the admission to a general hospital predated the diagnosis of intellectual disability in the SLaM record. However, we consider this number will be low as intellectual disability is a lifelong condition that is most often diagnosed in childhood, and the SLaM recording of intellectual disability in an adult is unlikely to be the first point at which the diagnosis is made.

Our analysis of factors associated with patient-level recording of intellectual disability is potentially affected by those who enter the study having less time to accrue admissions and recording accuracy changed over time. The consistency of findings in the sensitivity analysis helps to increase confidence in these results.

This study looks only at recording of intellectual disability in hospital records, using the patient discharge summary as the primary source of HES data. While helpful as a first step in contemplating and instituting reasonable adjustments, recording does not guarantee that adapted person-centred care will be provided, and we have no measure of the quality of care that was provided. Similarly, it is possible that in some cases where intellectual disability was not recorded, this was due to a failure of the administrative process alone, and that intellectual disability was indeed recognised and managed appropriately during the admission. Even in

these cases, however, it is still necessary from a service-level and surveillance perspective that intellectual disability is documented.

## Clinical and research implications

There is a need for improved recognition and recording of intellectual disability in general hospitals to improve hospital outcomes and care experiences. An active approach to identification has been suggested, with routine screening questions being asked at entry points to care [49]. Pre-admission assessments and learning disability identification checklists are recommended to ensure that people with intellectual disability receive the support they require in hospital and to give advance notice to staff [50,51] but are only possible in the case of planned admissions. People with intellectual disability are more likely than the general population to present to emergency care [52,53]; automatic flagging of people with intellectual disability attending hospital could be achieved with better integration of health records between statutory services, underpinned by stringent data sharing protocols and assurances about data confidentiality. A Canadian initiative demonstrated that linkage between health, education, and disability social care databases can capture a greater proportion and more varied group of people with intellectual disability and improve their visibility within statutory services, though the authors note that time and other resource constraints must be overcome as potential barriers to effective data linkage [54].

Recognition of intellectual disability should by law result in suitably adapted care in England, but does not necessarily guarantee a tailored care response, or indeed, result in better outcomes. It is important that awareness of intellectual disability is supplemented by all hospital staff having a broad understanding of the range of health and communication needs that people with intellectual disability may have and being familiar with general approaches to care. Access to specialists should then be available to support clinical teams to with more specific assessments and advice; this might include input from a learning disability liaison nurse or in-reach from a community learning disability team. Furthermore, there needs to be flexibility within the hospital ecosystem to enable truly individualised care, including extra time for interventions and allowing family members or carers to remain with the person where this may ordinarily not be permitted. Aside from these practical interventions, the presence of institutional discrimination against people with intellectual disability [16] demands a cultural shift that may be more difficult to effect and measure but it is hoped that a forthcoming programme of mandatory training in autism and learning disability for all NHS staff [55] will challenge misconceptions and contribute to improvements in care.

It is possible that "labelling" the intellectual disability may lead to stigma. A recent review of visual identifiers in the care of people with dementia highlighted potential ethical issues with flagging people with dementia in hospitals, but found that ethical and legal dilemmas could be overcome [56]. Alternatively, it would be possible to flag the domain(s) in which a person requires additional support, thereby labelling the adjustments rather than the disorder. This method could be applied to people with other deficits, such as those arising from dementia or sensory impairments; however, this would negate the ability to identify intellectual disability for legal compliance, research, and service planning purposes.

## Conclusions

This study demonstrates that, currently, HES data alone cannot be used to identify a cohort of people with intellectual disability attending English hospitals, as a significant proportion will be missed. Using a range of sources with health database linkage can increase coverage and

provide a powerful tool for epidemiological research to drive improvements in care in this group.

Future studies could investigate the experience and outcomes of care between those who were recorded as having intellectual disability during their admission and those who are not. Similar work could investigate recording of intellectual disability in out-patient clinics. Hospital recognition of other developmental disorders in whom health inequalities also exist, such as autism and borderline intellectual functioning, could also be investigated [57,58].

## Supporting information

**S1 Checklist. REporting of studies Conducted using Observational Routinely-collected Data (RECORD) guideline.**
(DOCX)

**S1 Table. ICD-10 codes for specific disorders almost always associated with intellectual disability.**
(DOCX)

**S2 Table. Time trends in recording intellectual disability in adults admitted to English general hospitals, 2005–2019.**
(DOCX)

**S3 Table. Odds of intellectual disability being unrecorded in the general hospital record of adults with intellectual disability attending hospital (with multiple imputation for missing data).**
(DOCX)

**S4 Table. Odds of intellectual disability being unrecorded in the general hospital record of adults with intellectual disability attending hospital (using a random effects model).**
(DOCX)

**S5 Table. Odds of intellectual disability being unrecorded in the general hospital record of adults with intellectual disability attending hospital (with multiple imputation for missing data).**
(DOCX)

**S1 Text. Study protocol.**
(DOCX)

## Author Contributions

**Conceptualization:** Rory Sheehan, Hassan Mansour, Angela Hassiotis, Christoph Mueller, Robert Stewart, Andre Strydom, Andrew Sommerlad.

**Data curation:** Rory Sheehan, Hassan Mansour, Matthew Broadbent, Christoph Mueller, Andrew Sommerlad.

**Formal analysis:** Rory Sheehan, Hassan Mansour, Matthew Broadbent, Christoph Mueller, Andrew Sommerlad.

**Methodology:** Rory Sheehan, Hassan Mansour, Matthew Broadbent, Angela Hassiotis, Christoph Mueller, Robert Stewart, Andre Strydom, Andrew Sommerlad.

**Project administration:** Rory Sheehan, Christoph Mueller, Andrew Sommerlad.

**Resources:** Robert Stewart.

**Supervision:** Andrew Sommerlad.

**Writing – original draft:** Rory Sheehan, Hassan Mansour, Matthew Broadbent, Angela Hassiotis, Christoph Mueller, Robert Stewart, Andre Strydom, Andrew Sommerlad.

**Writing – review & editing:** Rory Sheehan, Hassan Mansour, Matthew Broadbent, Angela Hassiotis, Christoph Mueller, Robert Stewart, Andre Strydom, Andrew Sommerlad.

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
