## [Editor Report · Decision Letter 0]

29 Sep 2022

Dear Dr Sheehan, 

Thank you for submitting your manuscript entitled "Recording of intellectual disability in general hospitals 2006-2019: cohort study using linked datasets" for consideration by PLOS Medicine.

Your manuscript has now been evaluated by the PLOS Medicine editorial staff as well as by an academic editor with relevant expertise and I am writing to let you know that we would like to send your submission out for external peer review.

Please re-submit your manuscript within two working days, i.e. by Oct 03 2022 11:59PM.

Kind regards,

Callam Davidson

Associate Editor

PLOS Medicine

---

## [Decision Letter · Decision Letter 1]

29 Nov 2022

Dear Dr. Sheehan,

Thank you very much for submitting your manuscript "Recording of intellectual disability in general hospitals 2006-2019: cohort study using linked datasets" (PMEDICINE-D-22-03204R1) for consideration at PLOS Medicine. 

Your paper was evaluated by an associate editor and discussed among all the editors here. It was also sent to independent reviewers, including a statistical reviewer. The reviews are appended at the bottom of this email and any accompanying reviewer attachments can be seen via the link below:

[LINK]

In light of these reviews, I am afraid that we will not be able to accept the manuscript for publication in the journal in its current form, but we would like to consider a revised version that addresses the reviewers' and editors' comments. Obviously we cannot make any decision about publication until we have seen the revised manuscript and your response, and we plan to seek re-review by one or more of the reviewers. 

We hope to receive your revised manuscript by Dec 20 2022 11:59PM. Please email us (plosmedicine@plos.org) if you have any questions or concerns.

We look forward to receiving your revised manuscript. 

Sincerely,

Callam Davidson, 

PLOS Medicine

plosmedicine.org

Please include the setting (England) in your title.

Abstract Methods and Findings:

* Please include the setting (England).

* Please include the important covariates that are adjusted for in the analyses.

Please ensure that the study is reported according to the RECORD guideline, and include the completed RECORD checklist as Supporting Information. Please add the following statement, or similar, to the Methods: "This study is reported as per the REporting of studies Conducted using Observational Routinely-collected Data (RECORD) guideline (S1 Checklist)."

The RECORD guideline can be found here: http://www.equator-network.org/reporting-guidelines/RECORD/

Please define the abbreviations in Table 1 in the legend.

Line 242: Please do not report P=0.0001; report as P < 0.001.

Please indicate in the figure caption the meaning of the bars in Figure 1.

Please define F70-79 and F81.9 in the legend of Figure 1.

Please clarify what is denoted by bold typeface in Table 3; if statistical significance, should univariate age also be in bold?

Please use et al. only after listing the first six authors in your References. See our website for other reference guidelines https://journals.plos.org/plosmedicine/s/submission-guidelines#loc-references

Comments from the reviewers:

Reviewer #1: This is a well-written paper about the recording of intellectual disabilities in hospital records from 2006 - 2019. This is a topic of importance, but I suggest that a more nuanced consideration of the issues is required before the paper is ready for publication. In particular, I suggest that the authors need to consider the following:

1. A definition of hospital admission is required, including how this might relate to hospital attendance.

2. A justification for why a deprivation index score has been included is needed. Many people with intellectual disabilities live in residential or supported living settings that bear no resemblance to socio-economic status of themselves or their families. 

3. A justification for the dates for inclusion in the study. This seems to me, to be a convenience sample, but the start date for inclusion, 2006, predates the requirements of the Equality Act 2010, which demands the provision of reasonable adjustments for disabled people. As such, including data before this legislation came into force, and making assumptions that it was already in force, creates a tension in the paper. 

4. More discussion about ICD10 coding is required, and the justification for only including the codes used. Glover, 2006 (file:///C:/Users/saph/Downloads/Howpeoplewithlearningdisabilitiesdie%20(1).pdf) includes a list in Appendix 1 of conditions always associated with intellectual disabilities, with the ICD10 codes. It would be instructive to understand the proportion of admissions with these codes but without F70-79.

5. The authors acknowledge that the rate of recording rose over time, being very different in 2019 from 2006. More recognition and discussion about this is needed, including suggestions as to the impact of the Equality Act, guidance for healthcare professionals (see for example, guidance for GPs and nursing staff), the introduction of learning disability nurses, findings from inquiries and reviews indicating the need for such recording over the years (e.g. CIPOLD study). A timeline, for example, illustrating the potential impact of such initiatives against the rise in reporting might be helpful. In addition, it might be helpful to present the findings from the most recent 2-3 years separately, so as to present a more contemporary picture of the situation.

6. The finding relating to married people needs to be treated with caution due to the small number. The reader needs to be reminded of this.

7. One of the aims of the study was to identify clinical and socio-demographic factors associated with the recording of intellectual disability. I am not sure what clinical factors have been identified; this needs to be clarified. 

8. A much broader discussion is required about whether or not a code for intellectual disability is likely to improve the issues related to poor quality care that are listed in the Introduction (lack of knowledge and understanding amongst health professionals, diagnostic overshadowing, institutional discrimination, inadequate communication and failure to acknowledge carers' expertise). There is possibly an argument to be heard that suggests that labelling a person with intellectual disabilities in this way, of itself leads to prejudiced or discriminatory care.

9. I would also have liked more consideration to be given as to whether coding intellectual disabilities is, in fact, more appropriate than coding the reasonable adjustments required by an individual. People with intellectual disabilities can have very different needs. Although there may be a 'menu' of reasonable adjustments that may be appropriate, what is important is the individual tailoring of these to the needs of an individual. A reflection on whether this is a better way forward than the rather blunt coding of 'intellectual disabilities' would be helpful.

10. Finally, an area of confusion for me is that in the section about the study limitations, it is noted that 'This study looks only at recording of intellectual disability in hospital records, using the patient discharge summary as the source of HES data'. This seems to contradict what is written in the Methodology section where it is stated that: 'We used HES inpatient admission data which include diagnoses identified during the hospital contact...' This requires clarifying.

Reviewer #2: Alex McConnachie, Statistical Review

Sheehan et al looks at the recording of intellectual disability in routine data for hospital admissions of people with ID in a large mental health trust in England between 2006 and 2019. This review considers the use of statistics in the paper.

Generally, these are quite well done. Outcomes are defined at the admission level and patient level, and are reported for all admissions and non-elective admissions. A strict coding definition and a wider definition are used. Logistic regression is used to assess the factors associated with poorer recording (at the patient level). Multiple imputation is also done, to assess the robustness of the main results. The interpretation of the statistical results are reasonable.

My main worry with the analysis is whether the data are suitable to answer the question. For someone with ID, is it necessarily the case that ID should be recorded in HES data? Are the diagnoses recorded in HES inpatient stay data more to do with the clinical reason for the admission? If the admission has nothing to do with the fact that the patient has ID, is there any reason for ID to be recorded in the HES data? The authors question whether the low level of recording is to do with administrative error, though my point is whether this is an "error" at all. It may be that a patient's ID is appropriately recognised during their admission, but unless there is some connection, this is not recorded as being a reason for their admission, or is only recorded occasionally.

That said, the results are still of interest. They show that HES data are not a reliable source of data about patients' ID status. They raise the question as to whether patients' ID status is appropriately recognised in secondary care, despite the low level of recording in routine data.

My remaining comments are minor. Some are simply suggestions for analyses that could be done, and probably reflect my biases as a reviewer. They might be of academic (statistical) interest, without adding much additional insight, or may be considered more complex than is warranted for a general medical audience. 

In the logistic regression models, the analysis is at the patient level, and the number of admissions each patient has is adjusted for. How good is this adjustment? Does the log odds of ID being recorded increase in a linear way with the number of admissions? Note that the number of admissions someone has is not a baseline characteristic - it is based on what happens to the patient after the baseline, which always has the potential to lead to bias. E.g., those people who enter the study cohort later on in the study period will have less time to accrue admissions, and therefore fewer opportunities for ID to be recorded, though the authors present analyses that suggest an increase in recording (through the F81.9 code) over time, so it is not clear how these competing factors will affect the patient-level recording of ID.

Was an admission-level analysis considered? This would avoid the problem noted above, but would have to account for clustering at the patient level. It could, however, assess whether recording is more likely during a non-elective admission, and whether there is a correlation over time in ID recording - i.e., once ID has been recorded during one admission, is it more likely to be recorded in subsequent admissions?

The authors suggest that it is not possible to measure quality of care. However, an analysis could be done looking at the time to the second (non-elective) hospital admission after the first. Whether or not ID was recorded during the first admission could be assessed as a covariate. Are patients who's ID status was recorded during their first admission (and therefore presumably appropriately recognised during that admission), at higher or lower risk of subsequent admissions? Such an analysis could even be done looking at the times from each admission to the next, though this would need to account for patient-level clustering again, and could become quite complex.

The discussion notes that a quarter of patients could have a genetic cause for ID, which could be recorded with different ICD codes. This seems like quite a large proportion. Would it be feasible to identify these patients (N ~ 600) and their genetic conditions, to see if these are recorded in the HES data?

In Table S1 (and similar), the title says "F70-79 or F81", but within the table, the term used is "F70-79 and F81", which is inconsistent.

Reviewer #3: This paper links two healthcare databases to assess the rate at which intellectual disability has been recorded in hospital records for patients an intellectual disability. This is an extremely well written paper on an important topic. I have only a few minor suggestions for the authors. 

* I would suggest the authors review some of the broader literature on recording of intellectual disability in health records. This can help demonstrate that this is a widespread issue and potentially support speculation on the challenges leading to poor documentation. For example see Lin et al. (2015) Strengths and Limitations of Health and Disability Support Administrative Databases for Population-Based Health Research in Intellectual and Developmental Disabilities

* I was surprised to see the much higher use of the F81.9 code. Do the authors have any theories on why this was used at such higher rates and whether there are any clinical consequences of using this code?

Reviewer #4: The paper entitled "Recording of intellectual disability in general hospitals 2006-2016: cohort study linked datasets" by Sheehan R et al. aimed to 1) investigate the recording of intellectual disability in adults with a confirmed intellectual disability diagnosis who were admitted to general hospitals, 2) analyse changes in the recording of intellectual disability in those admitted to hospital over time and 3) identify clinical and socio-demographic factors associated with intellectual disability being unrecorded in those with the condition. 

Overall comment:

In general, this manuscript is well written and well organised to be accessible to non-specialists. However, overall I felt the paper did not elaborate enough on the importance of intellectual disability being recognised in general hospitals. For example, the authors may have benefited from comparing the average length of stay of episodes where intellectual disability is being recognised versus unrecognised. The investigation could highlight and support their claim that recognising intellectual disability is essential for providing additional support and reasonable adjustments in this population. I also have some concerns around the methodology used, highlighted below.

The authors might not be aware of one article that examined factors associated with recognising intellectual disability for adults during hospital admission in Australia (Walker AR, Trollor JN, Florio T, Srasuebkul P. Predictors and outcomes of recognition of intellectual disability for adults during hospital admissions: A retrospective data linkage study in NSW, Australia. PLoS One. 2022 Mar 25;17(3):e0266051. doi: 10.1371/journal.pone.0266051. PMID: 35333913; PMCID: PMC8956190.). 

Methods

I have multiple concerns with the method that I believe need to be addressed before publication. 

1. Although the authors used strict ICD10 codes to identify intellectual disability, I recommend the codes be broadened to include some conditions like Down Syndrome or other congenital malformations that are linked to intellectual disability.

2. I found that the outcome, which is the failure to recognise intellectual disability, is unintuitive. When the outcome is framed this way (rather than as the positive recognition of intellectual disability), most records have an outcome. In general, logistic regression can perform poorly when a large majority of records have positive outcomes. It also makes it unintuitive for readers to understand the results, as it is predicting the lack of an event. I recommend that the authors change the outcome to intellectual disability being recognised. 

3. I liked having a gold standard and comparing it with another dataset. However, have the authors considered that staff at general hospitals might not know that someone had an intellectual disability before the diagnosis was recorded in the SLaM BRC? Instead of using all HES episodes, authors should use hospital episodes after the formal diagnosis has been recorded in SLaM.

4. I don't think regular logistic regression is appropriate; authors should consider a random effect model which controls for within-person effect or a generalised estimating equation. 

5. The use of the term 'sensitivity' is an interesting choice here. For statisticians, the term 'sensitivity' refers to a given test's ability to correctly designate an individual with a disease (or other binary outcomes) as positive. In this context, I think what the authors presented is the proportion of episodes or people that are correctly identified as having an intellectual disability. 

Reviewer #5: 

Title: Recording of intellectual disability in general hospitals 2006-2019: cohort study using linked datasets

This linked data study utilised information from a large secondary mental healthcare database to identify those with intellectual disability and linked these individuals to general hospital records, to investigate how well intellectual disability had been recorded in medical files. I assume therefore that the cohort will be restricted to those individuals with intellectual disability and co-morbid mental health conditions?. This may need to be clarified. The study aimed to look at trends over time and factors associated with the non-recording of intellectual disability.

Introduction

The introduction has provided evidence of the high morbidity and earlier mortality experienced by individuals with intellectual disability. They also experience a higher number of hospitalisations than those without intellectual disability as has been evidenced across multiple countries. The authors state little information exists on how well intellectual disability is recognised and recorded in general hospital settings and may not be aware of an Australian study which has also investigated how well individuals with intellectual disability were coded as such in hospital records. This study showed only about 15% received an intellectual disability code although it used a younger cohort. (Ref Bourke J, Wong K, Leonard H. Validation of intellectual disability coding through hospital morbidity records using an intellectual disability population-based database in Western Australia. BMJ open. 2018 Jan 1;8(1):e019113.)

Methods

Methods seem appropriate and well-explained. The authors used only ICD codes specific to intellectual disability (F70-79) to identify ID and did not include codes for conditions that are consistent with intellectual disability such as Down syndrome, Fragile X or Prader-Willi. Although some conditions may not be 100% associated with ID this may have explained some of the un-coded ID- particularly for those with Down syndrome. Coding practices may not see the need for an intellectual disability code in those with Down syndrome? I see that this issue has been addressed in the discussion.

Results

Table 2 would probably be clearer if the row labelled 'ICD_10 Codes' was simply included in the row above it and labelled 'ICD-10 Descriptor and codes'.

The increase in % identified through ICD codes when F81.9 is included (Table 2) possibly indicates the large proportion of individuals who may have borderline intellectual disability and there is not the confidence to assign a specific level of ID using the F70-79 codes, even though they do include an unspecified level, F79. The time trend (Figure 1) is also interesting to see this code obviously being used more frequently over time.

Discussion

The discussion makes many good points justifying the need for correct coding of intellectual disability in the hospital setting, particularly to aid in communication during treatment and discharge. It is also important to have this information recorded for use in resource planning. 

The study is clearly written but does have some limitations in being limited to those individuals in the catchment area of the mental health services. However the results are very useful and an important point made is the limitation in using only hospital coding to identify a cohort of individuals with intellectual disability for any research purpose.

Minor typos

Page7, line 130: …diagnosis of intellectual disability (and) was taken as either a record of an ICD-10 cod

Page10, line 203: The sample comprised 2,477 adults with intellectual disability who were admitted (to) a general hospital

[LINK]

---

## [Decision Letter · Decision Letter 2]

7 Feb 2023

Dear Dr. Sheehan,

Thank you very much for re-submitting your manuscript "Recording of intellectual disability in general hospitals in England 2006-2019: cohort study using linked datasets" (PMEDICINE-D-22-03204R2) for review by PLOS Medicine.

I have discussed the paper with my colleagues and the academic editor and it was also seen again by three reviewers. I am pleased to say that provided the remaining editorial and production issues are dealt with we are planning to accept the paper for publication in the journal.

[LINK]

We look forward to receiving the revised manuscript by Feb 14 2023 11:59PM.   

Sincerely,

Callam Davidson

Associate Editor

PLOS Medicine

plosmedicine.org

Requests from Editors:

Abstract Methods and Findings:

* Please ensure that all numbers presented in the abstract are present and identical to numbers presented in the main manuscript text.

Did your study have a prospective protocol or analysis plan? Please state this (either way) early in the Methods section.

Comments from Reviewers:

Reviewer #1: Thank you to the authors for their consideration of the reviewers comments and for the clear and transparent way in which they have made changes to this paper in line with the comments.

In my view, the paper is now ready for publication.

Reviewer #2: Alex McConnachie, Statistical Review

I thank the authors for their consideration of my original comments.

Knowing that the number of admissions is included in the models as a categorical variable is useful, but it would be of interest to see the odds ratios associated with this factor in the adjusted regression models. However, I note that this adjustment is included in the adjusted mixed effects model at the admission level. I am not sure that this makes sense, since the unit of analysis is the admission; why should the total number of admissions be associated with recording within any one admission?

It is good to see the admission-level mixed effects model results. These raise a few questions. The age association is reversed - in the main analysis, older age is associated with an increased odds of non-recording, but in the mixed effects analysis, it is associated with a reduced odds of non-recording. This needs to be explained. Also, in the mixed model, I assume that the age variable represents age at the time of the admission, not at baseline?

The discussion, around lines 424-428, says that factors other than degree of intellectual disability and marital status were not associated with recording, but in the mixed model, both ethnicity and IMD decile showed evidence of associations. I think the differences between the patient-level and admission-level regression analyses need to be discussed more.

There is a statement around line 430 that recording in elective and non-elective admissions was similar, though I do not recall seeing this tested anywhere. Data are reported for all admissions, and for non-elective admissions, but not for elective admissions, specifically. From the data presented, it appears that recording during non-elective admissions is lower than during elective admissions, if using the strict coding definition, but is higher when including the F81.9 code. The mixed model would be the ideal place to assess this - elective or non-elective could easily be added as a covariate.

Whilst I am not insisting the analysis be done, I think the response to the suggestion of a time to event analysis did miss the point a little. I specifically suggested that the time to non-elective admission be analysed, so the fact that people with ID require regular admissions is perhaps less relevant. Also, with a time to event (i.e. survival) analysis, not having the event of interest is not a barrier to including people in the analysis.

Reviewer #4: The authors did a great job responding to all queries. I do not have any more comments.

[LINK]

---

## [Decision Letter · Decision Letter 3]

20 Feb 2023

Dear Dr Sheehan, 

On behalf of my colleagues and the Academic Editor, Professor Pauline Heslop, I am pleased to inform you that we have agreed to publish your manuscript "Recording of intellectual disability in general hospitals in England 2006-2019: cohort study using linked datasets" (PMEDICINE-D-22-03204R3) in PLOS Medicine.

When making the formatting changes, please also address the following editorial comment:

* Line 48: Please replace the word 'higher' with 'specific', or a similarly appropriate alternative.

PRESS

Sincerely, 

Callam Davidson 

Associate Editor 

PLOS Medicine